# Dissipation Dynamics and Dietary Risk Assessment of Four Fungicides as Preservatives in Pear

Yongfeng Tang [1], Kuikui Hu [1], Xiaomeng Li [1], Chaogang Liu [2], Yanhui Xu [3], Zhaoxian Zhang [1] and Xiangwei Wu [1,*]

[1] Key Laboratory of Agri-Food Safety of Anhui Province, College of Resources and Environment, Anhui Agricultural University, Hefei 230036, China; tyf@ahau.edu.cn (Y.T.); 19722370@stu.ahau.edu.cn (K.H.); 20721615@stu.ahau.edu.cn (X.L.); zhangzx@ahau.edu.cn (Z.Z.)

[2] Dangshan Sanlian Fruit and Vegetable Professional Cooperative, Dangshan 235300, China; liuchaogang@ahau.edu.cn

[3] Anhui Public Inspection Institute Co., Ltd., Hefei 230051, China; xuyanhui@ahau.edu.cn

* Correspondence: wxw@ahau.edu.cn; Tel./Fax: +86-551-65786320

**Abstract:** Fungicides, including thiophanate-methyl, tebuconazole, pyraclostrobin, and difenoconazole, have been widely used as preservatives to control fungal diseases during pear storage. However, the metabolic capability of pear for exogenous compounds decreases at lower storage temperatures, leading to an increase in the risk of exposure to chemical preservatives. In this work, a sensitive and stable ultraperformance liquid chromatography–tandem mass spectrometry (UPLC–MS/MS) analytical method was established to investigate the dissipation dynamics and dietary intake risk of four chemical preservatives in pears under different conditions. The mean recoveries of the preservatives in pear samples ranged from 73.2% to 117.1%, with relative standard deviations of 0.5–7.2%. The dissipation half-lives ($T_{1/2}$) of thiophanate-methyl, tebuconazole, pyraclostrobin, and difenoconazole in pears were 7.2–21.1 d and 31.6–173.3 d at storage temperatures of 25 °C and 4 °C, respectively. The results of dietary risk evaluation showed that the intake risk of preservatives in commercial pears was acceptable. However, some pears from commercial supermarkets still contained preservatives at amounts that exceeded the maximum residue limit (MRL) set by the Chinese government. This work provides a guideline for the risk evaluation of fruit preservatives on human health.

**Keywords:** fungicides; pear; dissipation dynamics; pesticide residues; dietary risk assessment

## 1. Introduction

With massive amounts of vitamins, potassium, and calcium, pear is one of the main fruit varieties in the world [1,2]. Pear possesses therapeutic effects on some diseases, including hypertension and cardiopathy, and prevents laryngeal, lung, and nasopharyngeal cancer [3–5]. However, pears are susceptible to infection by pathogenic fungi and bacteria during storage, which causes deterioration [6,7]. Furthermore, the pathogenic fungi in pears can produce some mycotoxins that affect human health [8,9]. Currently, chemical preservatives are an important tool to prevent the rotting of fruits during storage [10,11]. Fungicides, including tebuconazole, carbendazim, thiophanate methyl, mancozeb, difenoconazole, pyraclostrobin, and prochloraz, are commonly used in pears as chemical preservatives [12,13]. Although chemical preservatives exhibit good effects in controlling pathogenic microorganisms, most preservatives have varying degrees of toxicity and residues that cause acute and chronic toxicity [14]. Freitas et al. found that tebuconazole exposure could induce an increase in transaminase and serum testosterone levels in fruit bats, leading to an endocrine disorder [15]. Yang et al. investigated the potential toxicity of tebuconazole to male rats by chronic exposure at environmental concentration levels, and the results suggested that tebuconazole could decrease the concentration of serum testosterone and cauda epididymal sperm count and cause antiandrogenic activity [16]. Pyraclostrobin led to adverse health effects on mice, including weight loss, hypothermia, and diarrhea [17]. Zhang et al. studied

the toxicity of pyraclostrobin in zebrafish and found that it could cause DNA damage and inhibit the activity of antioxidant enzymes [18]. Difenoconazole had toxic effects on zebrafish, including hatching inhibition, abnormal spontaneous movement, slow heart rate, growth regression, and morphological deformities [19]. Jia et al. demonstrated that thiophanate methyl was rapidly degraded and transformed to the metabolite carbendazim [20]. The fungicide thiophanate methyl specifically induced serious hepatotoxicity in zebrafish larvae and adults. Thus, some chemical fungicides, such as captan and benomyl, have been banned by the United States Environmental Protection Agency (EPA) for fruit storage.

The degradation rate of exogenous compounds in fruits is reduced because of the decreased metabolic enzyme activity during fruit storage at low temperatures. The application of chemical fungicides as preservatives can increase the probability of excessive pesticide residues. Fang et al. investigated the dissipation behavior of prochloraz, pyraclostrobin, and tebuconazole in pears stored under different conditions using ultraperformance liquid chromatography. The results showed that the $T_{1/2}$ ranges for degradation of the three fungicides in pear peel were 8.8–13.9 d after storage at 25 °C and 99.0–346.6 d after storage at 2 °C. Under 2 °C storage conditions, the maximum residual concentrations of the prochloraz, pyraclostrobin, and tebuconazole in pears were 0.363, 1.871, and 0.226 mg/kg after 180 d [13].

In the present study, a sensitive and stable multiple residue analysis method was established with ultraperformance liquid chromatography–triple quadrupole mass spectrometry (UPLC–MS/MS) for the simultaneous determination of tebuconazole, pyraclostrobin, difenoconazole, thiophanate-methyl, and carbendazim (a major metabolite of thiophanate-methyl) in pears. The dissipation dynamics of the four preservatives applied by spray to the pears were investigated at storage temperatures of 25 °C and 4 °C. Furthermore, the marker samples of pears were collected from different supermarkets to measure the residual amounts of the preservatives, and a dietary risk evaluation was carried out based on the residual data. The results provide a scientific basis for rationalizing the use of chemical preservatives and improving the quality safety of pears.

## 2. Materials and Methods

### 2.1. Chemicals and Reagents

The standards carbendazim (98.6% purity), tebuconazole (98.6% purity), thiophanate-methyl (98.9% purity), pyraclostrobin (99.5% purity), and difenoconazole (99.58% purity) were obtained from Dr. Ehrenstorfer GmbH. The commercial pesticide preparations of 70% thiophanate-methyl wettable powder (WP), 40% tebuconazole suspension concentrate (SC), 30% pyraclostrobin SC, and 40% difenoconazole SC used for field experiments were purchased from Shandong Kangqiao Biotech. Co., Ltd. (Qingdao, China). Methanol (MeOH) and acetonitrile (ACN) of chromatographic grade were purchased from Sigma-Aldrich Co. LLC (Shanghai, China). Methanoic acid was provided by Tianjin Guangfu Fine Chem. Co., Ltd. (Tianjin, China). A solid-phase extraction $NH_2$ column (500 mg/6 mL) (SPE-$NH_2$) was obtained from Agela Technologies (Tianjin, China). Other analytical- or chromatographic-grade reagents were purchased from InterBusiness Co. Ltd.

### 2.2. Analytical Methods for the Determination of the Four Fungicides in Pears

2.2.1. Sample Preparation

An aliquot of 5.0 g homogenized pear was accurately weighed into a 50 mL polytetrafluoroethylene centrifuge tube with 5.0 g NaCl and 20 mL of ACN. The mixtures were vortexed for 30 min on a vortex mixer (VXMTA, Ohaus, Parsippany, NJ, USA). The tube was centrifuged for 5 min at $3913 \times g$, and 2 mL of the supernatant was subsequently collected for further purification. An SPE-$NH_2$ column was used to clean the extracted samples. First, the SPE-$NH_2$ column was rinsed with 5 mL of ACN, and then 2 mL of supernatant was transferred to the column, and the eluate was collected. The SPE-$NH_2$ column was further eluted with 2 mL of ACN. Finally, all the eluates were combined and concentrated to dryness with a nitrogen stream on a pressure-blowing concentrator (Ruicheng Instrument

Co., Ltd., Hangzhou, China). The extracted residues were dissolved in 2 mL of ACN and passed through a 0.22 μm nylon syringe filter before UPLC–MS/MS analysis.

### 2.2.2. Determination of the Four Fungicides

The fungicides carbendazim, tebuconazole, thiophanate-methyl, pyraclostrobin, and difenoconazole were quantitatively and qualitatively analyzed using an ACQUITY ultraperformance liquid chromatograph combined with an XEVO triple-quadrupole mass spectrometer (UPLC–MS/MS) (Waters Crop., Milford, CT, USA) with an ACQUITY UPLC BEH C18 column (100 mm × 2.1 mm, 1.7 μm, Waters Crop.) at a flow rate of 0.3 mL/min, and column temperature of 35 °C. The injection was 5 μL. The mobile phase consisted of 0.1% methanoic acid aqueous solution with 2% MeOH (A) and ACN with 0.1% methanoic acid (B). The gradient elution procedure is shown in Table S1.

Mass spectrometry was performed with a positive electrospray ionization (ESI+) source in multiple reaction monitoring (MRM) mode. The MRM transitions and collision energies were optimized during the experiments, and the optimal quantitative and quantitative ions are shown in Table S2. Other MS/MS conditions were as follows: capillary voltage, 3.0 kV; desolvation temperature, 350 °C; desolvation gas flow, 650 L/h; cone gas flow, 50 L/h.

### 2.2.3. Method Validation

The multiple residue analysis method was evaluated using the specificity, linearity, limit of detection (LOD), limit of quantification (LOQ), accuracy, and precision. Blank pear samples without carbendazim, tebuconazole, thiophanate-methyl, pyraclostrobin, or difenoconazole were collected from a pear orchard in Dangshan, Anhui Province, China. The linearity of solvent and matrix-matched calibration curves were determined at concentrations of 1, 2.5, 5, 10, 25, 50, and 100 μg/L. The matrix standard sample was diluted with extracting solution of blank pear sample. The slope ratio of the solvent and matrix-matched calibration curves was calculated to evaluate the matrix effect. The matrix-dependent LODs and LOQs of carbendazim, tebuconazole, thiophanate-methyl, pyraclostrobin, and difenoconazole were defined as the lowest concentrations that produced 3- and 10-fold signal-to-noise (S/N) ratios, respectively, to characterize the sensitivity of the method. The accuracy and precision of the method were examined by fortified recovery experiments. The pear samples in quintuplicate were spiked with 100 μL of the mixed standard solution with concentrations of 0.5, 5, 50, 500, and 2500 mg/L, and the final concentrations were 0.01, 0.1, 1, 10, and 50 mg/kg. The target analytes were extracted according to the sample preparation process described in Section 2.2.1. The recoveries and relative standard deviations (RSDs) were calculated to evaluate the analysis method.

### 2.3. Field Trials

The field trials were conducted on Aug 31, 2021 at Dangshan, Anhui Province, China. The meteorological conditions before and during sampling are shown in Table S3. A total of 12.5 L of dilutions (800- and 1200-fold dilutions of 70% thiophanate-methyl WP, 4000- and 5000-fold dilutions of 40% tebuconazole SC, 3200- and 4000-fold dilutions of 30% pyraclostrobin SC, and 1200- and 2400-fold dilutions of 40% difenoconazole SC) was sprayed on pear trees based on the recommended doses of these fungicides in triplicate. Two hours after spraying, the pear samples were collected and stored at 4 °C and 25 °C to evaluate the dissipation rate of the preservatives at different storage temperatures [13,21,22]. Samples were collected at 0, 1, 5, 10, 15, 20, 30, and 45 d at 25 °C and 0, 10, 20, 30, 45, 60, 75, and 90 d at 4 °C for the analysis of preservative residues.

The dissipation dynamics of the fungicides were described with the first-order kinetic Equation (1):

$$C_t = C_0 \times e^{-kt} \tag{1}$$

where $C_t$ is the concentration of tebuconazole, thiophanate-methyl, pyraclostrobin, or difenoconazole at time $t$ (mg/L); $C_0$ is the initial concentration; and $k$ is the dissipation rate constant.

The half-life ($T_{1/2}$) of the fungicides was calculated with Equation (2):

$$T_{1/2} = In2/k \qquad (2)$$

### 2.4. Dietary Risk Evaluation

To evaluate the intake risks of the chemical fungicides carbendazim, tebuconazole, thiophanate-methyl, pyraclostrobin, and difenoconazole, commercial pear samples were purchased from the production regions (Dangshan, Anhui Province) and different supermarkets in Hefei city, Bozhou city, and Bengbu city, which are located in Anhui Province, China, to determine the levels of five fungicide residues. The dietary risk assessment was carried out via three methods.

Method 1: The national estimated daily intake (*NEDI*) for long-term intake risk and the risk quotient (*RQ*) were calculated for dietary risk evaluation using Equations (3) and (4), respectively [23–25]:

$$NEDI = \sum \frac{STMR_i}{STMR - P_i} \times F_i \qquad (3)$$

$$RQ = \frac{NEDI}{ADI} \times bw \qquad (4)$$

Method 2: The dietary risk assessment (chronic risk) intake (*IEDI*, mg/kg bw) and risk quotient (*RQ_c*) for each fungicide were calculated using Equations (5) and (6), respectively, for long-term dietary risk assessment (chronic risk) [26]:

$$IEDI = STMRi \times F_i / bw \qquad (5)$$

$$RQ_C = IEDI / ADI \qquad (6)$$

Method 3: The acute dietary risk (*RQa*) was assessed using the short-term intake (*NESTI*). These were calculated by Equations (7) and (8), respectively [27]:

$$NESTI = Fi \times HR / bw \qquad (7)$$

$$RQa = NESTI / ARfD \qquad (8)$$

where *STMRi* (mg/kg) is the median residue of the fungicides from experimental market samples or in pears, $STMR\text{-}P_i$ is the median residue corrected by the correction factor, $F_i$ is the food consumption of Chinese people (kg), and *ADI* is the acceptable daily intake (mg/kg bw). Generally, the *RQ*, *RQ_c*, and *RQa* values were less than 100%, suggesting that the risk was acceptable for common consumers.

### 2.5. Data Analysis

The data were analyzed using the Origin 8.5 software (OriginLab Corp., Northampton, MA, USA). The variables studied in all experiments are expressed as the mean $\pm$ SD (standard deviation) of at least three independent assays, with each assay repeated 3–5 times. When comparing different groups of treatment and control, a parametric test of ANOVA by SPSS Statistics Software 22.0 (IBM Corp., Armonk, NY, USA) was used to evaluate the significance level ($p < 0.05$).

## 3. Results

### 3.1. Optimization of the Extraction and Cleanup Procedure

Extractants and cleanup sorbents play an important role in the efficiency of extraction and cleanup [28]. The extraction solvent was the main factor affecting the extraction efficiency. Organic solvents such as ACN, MeOH, and acetone are commonly used as extraction solvents. An appropriate amount of acid mixed with the extraction solvent

might improve the extraction efficiency [29,30]. The extraction efficiency of five fungicides in pear samples was evaluated using four organic solvents, MeOH, MeOH with 0.5% formic acid, ACN, and ACN with 0.5% formic acid. The results showed that the recovery rates of the five fungicides except for pyraclostrobin were less than 50% when MeOH or MeOH with 0.5% formic acid was used as the extraction solvent, indicating that MeOH was not suitable for the extraction of the five fungicides (Figure 1A). For ACN or ACN with 0.5% formic acid, the recovery rates were greater than 90%, which met the requirements of residual analysis of pesticides. No significant difference in the extraction efficiencies for thiophanate-methyl, pyraclostrobin, or difenoconazole was observed between ACN and ACN with 0.5% formic acid as the extraction solvent, while the extraction efficiencies for carbendazim and tebuconazole using ACN with 0.5% formic acid were superior to those of ACN. These results indicated that ACN with 0.5% formic acid was the most suitable solvent for the extraction of the five fungicides in pears. This might be related to the better penetrability and lower lipophilic properties of ACN.

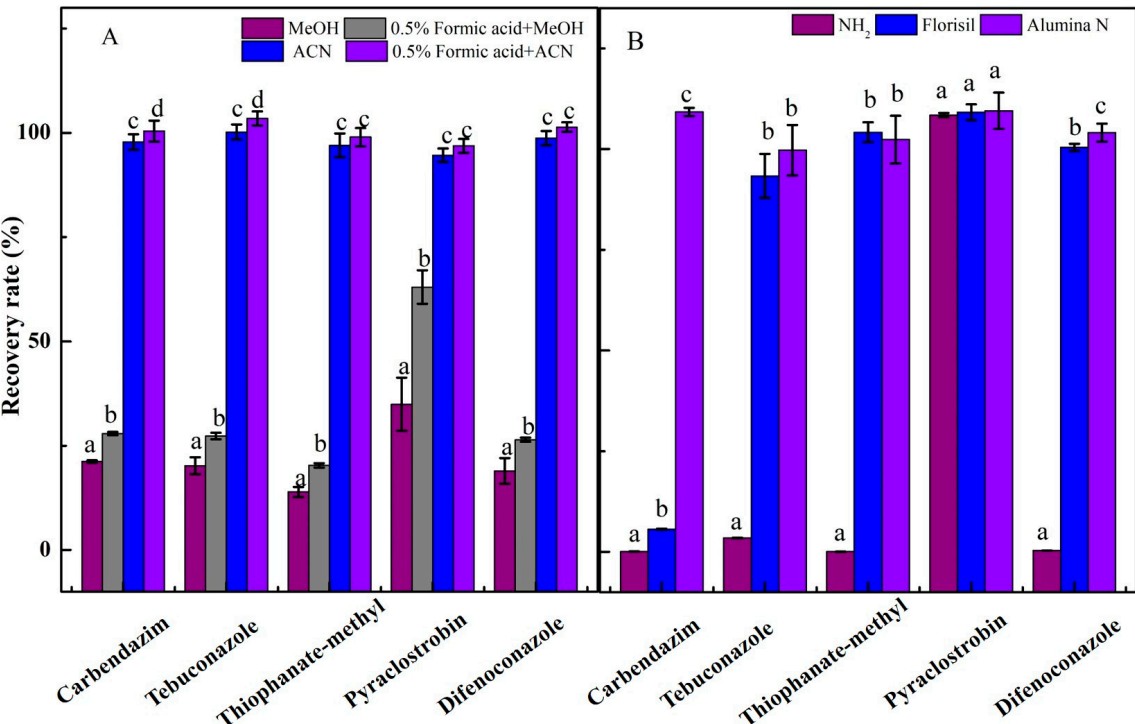

**Figure 1.** Effect of the different extraction solvents (**A**) and solid-phase extraction columns (**B**) on the extraction and purification of pear samples, respectively. The different lowercase letters are significantly different at $p < 0.05$.

Solid-phase extraction columns with various sorbents are generally used to purify the extraction solution to obtain satisfactory recovery of the target analytes [28]. In the present study, the effects of different solid-phase extraction columns, including Cleanert®NH$_2$, Cleanert®Florisil, and Cleanert®Alumina N, on the extraction efficiency were evaluated (Figure 1B). Only the pyraclostrobin recovery in the pear sample was satisfactory (107.6–109.0%) when the neutral alumina column was employed (Figure 1B). Compared with the alumina column, the Florisil column had better recoveries (89.1–111.8%) for tebuconazole, thiophanate-methyl, pyraclostrobin, and difenoconazole in the pear sample. However, the recovery of carbendazim in the pear sample was less than 10% using the Florisil column. To achieve good efficiency, the amino solid-phase extraction column resulted in good recoveries (92.0–116.7%) with RSDs of 1.0–6.3% for all five fungicides. Therefore, the NH$_2$ solid-phase extraction column was selected as the cleanup column for the pear sample.

### 3.2. Method Validation

The specificity analysis showed no endogenous compound interference around the retention time of the five target analytes (Figure 2). As shown in Table 1, satisfactory linearity for solvent and matrix matched calibration curves was obtained with a correlation coefficient ($R^2 > 0.999$). Coeluting components of the analytes resulted in matrix-enhanced or matrix-inhibited effects, which affected the ionization efficiency in UPLC–MS/MS, leading to quantitative inaccuracy [28]. The slope ratio of the matrix/solvent was far less than 1, which meant that there was a matrix inhibition effect. Therefore, the matrix-matched standard curve was used to quantify the five fungicides in pears. The matrix-matched LODs and LOQs were 0.16–2.5 ng/kg and 0.60–8.4 ng/kg, respectively (Table 1). The results of the spiked recovery experiments showed that the recoveries of the five fungicides in pears ranged from 73.2 to 117.1%, with RSDs of 0.5–7.2% (Table S4). Therefore, the analysis method was suitable for the determination of fungicide residues in pears.

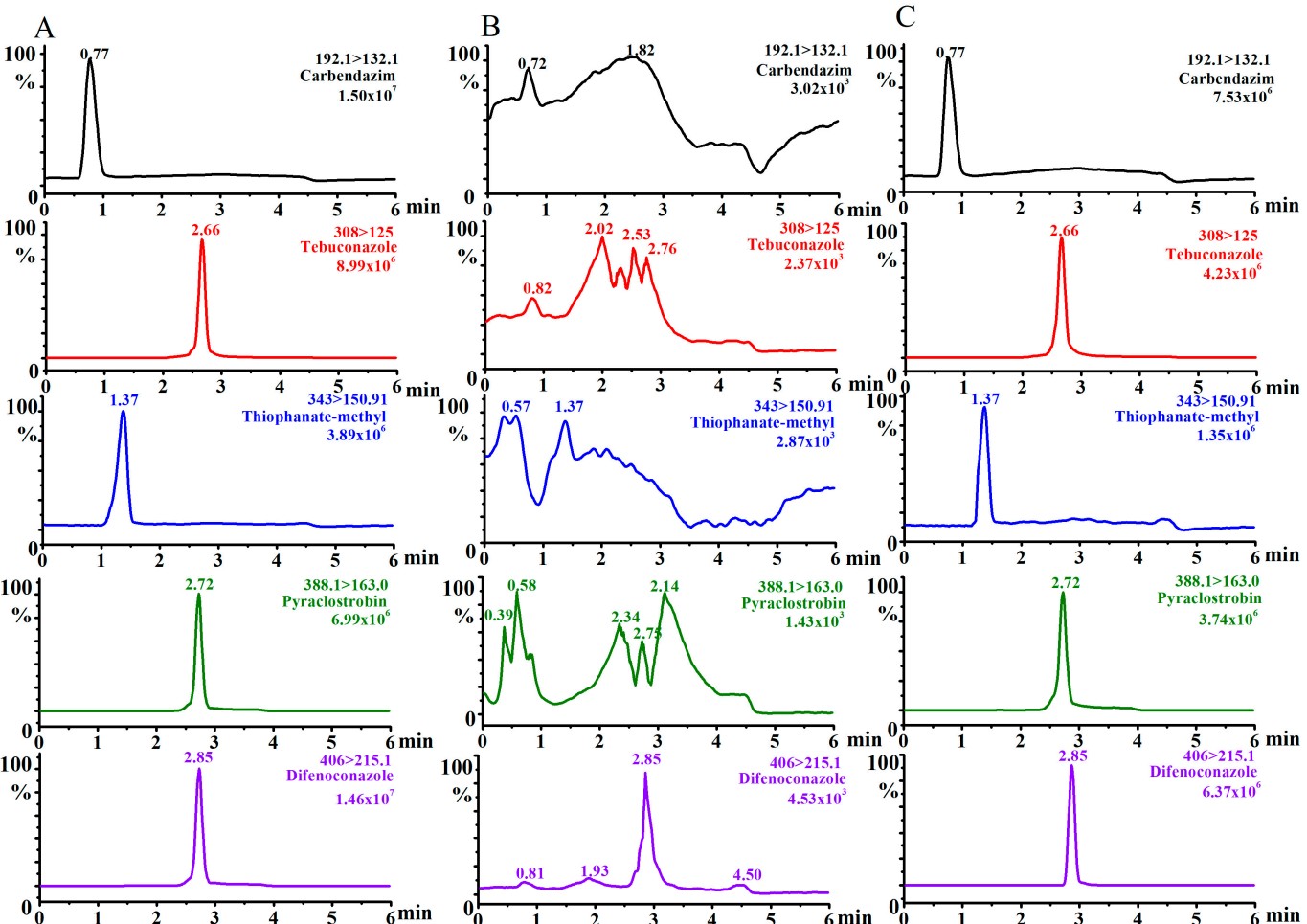

**Figure 2.** Typical UPLC–MS/MS multiple reaction monitoring chromatograms of the five fungicides from the standard solution (0.1 mg/L) (**A**), blank pear sample (**B**), and spiked pear sample (1 mg/kg) (**C**).

**Table 1.** Linear ranges, matrix effects, limits of detection (LODs), and limits of quantitation (LOQs) of the five fungicides.

| Matrix | Compounds | Liner Range (mg/kg) | Calibration Curves | Correlation Coefficients | Slop Ration | ME (%) | LOD (ng/kg) | LOQ (ng/kg) |
|---|---|---|---|---|---|---|---|---|
| Acetonitrile | Carbendazim | 0.001–0.1 | Y = 105045X − 31881 | 0.9997 | / | / | / | / |
| | Tebuconazole | 0.001–0.1 | Y = 110317X + 50545 | 0.9993 | / | / | / | / |
| | Thiophanate-methyl | 0.001–0.1 | Y = 34102X − 10859 | 0.9996 | / | / | / | / |
| | Pyraclostrobin | 0.001–0.1 | Y = 88561X + 157793 | 0.9991 | / | / | / | / |
| | Difenoconazole | 0.001–0.1 | Y = 84024X + 54151 | 0.9994 | / | / | / | / |
| Pear | Carbendazim | 0.001–0.1 | Y = 4654.6X + 49335 | 0.9992 | 0.044 | 95.57 | 1.5 | 5.1 |
| | Tebuconazole | 0.001–0.1 | Y = 3898.9X + 14377 | 0.9992 | 0.035 | 96.47 | 0.16 | 0.6 |
| | Thiophanate-methyl | 0.001–0.1 | Y = 3157.8X + 34032 | 0.9993 | 0.093 | 90.74 | 2.5 | 8.4 |
| | Pyraclostrobin | 0.001–0.1 | Y = 3381.1X + 16999 | 0.9994 | 0.038 | 96.18 | 0.24 | 0.8 |
| | Difenoconazole | 0.001–0.1 | Y = 5667.4X + 4252.6 | 0.9990 | 0.067 | 93.26 | 2.0 | 6.5 |

Note: ME (%) = ((slope (matrix match)/(slope (solvent curve) − 1) × 100. Slope ratio = matrix/acetonitrile.

*3.3. Dissipation Behaviors of the Fungicides in Pears*

The dissipation dynamics of the fungicides as storage preservatives were investigated in pears. The dissipation trend of the five fungicides applied at two dosages conformed to a first-order kinetic equation, with $R^2$ values of 0.8147–0.9913 at storage temperatures of 4 °C and 25 °C (Table 2). For thiophanate-methyl, the $T_{1/2}$ was in the range of 7.2–41.3 d. The metabolite carbendazim was rapidly formed along with the degradation of thiophanate-methyl during storage (Figures 3 and 4). The concentration of carbendazim from the metabolism of thiophanate-methyl achieved its maximum (0.72 mg/kg) at 0 d under the storage condition of 25 °C at both application dosages, while the peak concentration of carbendazim appeared at 10 d in storage at 4 4 °C. This indicated that thiophanate-methyl dissipated slowly under storage at low temperatures. The $T_{1/2}$ of tebuconazole in 4 °C storage was 7.57–8.35 times slower than that in 25 °C storage at both application rates. For difenoconazole, the $T_{1/2}$ values were 7.5–21.1 d and 44.0–135.1 d in 25 °C and 4 °C storage conditions, respectively. Pyraclostrobin exhibited the longest $T_{1/2}$ of 173.3 d in 4 °C storage compared with the other fungicides. Based on the above results, it can be concluded that the $T_{1/2}$ of the four fungicides used as storage preservatives at 4 °C were significantly longer than those at 25 °C. Therefore, the exposure risk of chemical fungicides increases when they are used as the storage preservatives at low storage temperatures. Soliman et al. reported that the $T_{1/2}$ of thiophanate-methyl was 2.9–3.2 d, 2.5 d, and 2.6 d in strawberry, mango, and grape, respectively, under field conditions [31]. The $T_{1/2}$ of tebuconazole was 6.5–24.7 d in grape. The $T_{1/2}$ of pyraclostrobin was 6.1–12.7 d, 5.1–5.7 d, and 15.1–19.8 d in jujube, pepper, and apple, respectively [21,32–35]. The $T_{1/2}$ of difenoconazole was 3.4 d in *Lycium barbarum* [21]. Compared with the dissipation of the four fungicides when used in the field, thiophanate-methyl, tebuconazole, pyraclostrobin, and difenoconazole dissipated slowly indoors, leading to an increase in the exposure risk to human health.

**Table 2.** Dissipation dynamic parameters of four fungicides and half-lives ($T_{1/2}$) of the fungicides in pears at 25 °C and 4 °C storage conditions at two application dosages.

| Sample | Temperature (°C) | Dilution Ratio | Dynamic Equation | $T_{1/2}$ (d) | $R^2$ |
|---|---|---|---|---|---|
| Thiophanate-methyl | 25 | 1200-fold | $C_t = 0.2193 \times e^{-0.0626t}$ | 11.1 | 0.9542 |
| | | 800-fold | $C_t = 0.6472 \times e^{-0.0960t}$ | 7.2 | 0.9381 |
| | 4 | 1200-fold | $C_t = 0.5256 \times e^{-0.0168t}$ | 41.3 | 0.9570 |
| | | 800-fold | $C_t = 0.8516 \times e^{-0.0219t}$ | 31.6 | 0.9856 |
| Tebuconazole | 25 | 4000-fold | $C_t = 0.4000 \times e^{-0.0492t}$ | 14.1 | 0.9913 |
| | | 2000-fold | $C_t = 0.5474 \times e^{-0.0451t}$ | 15.4 | 0.9730 |
| | 4 | 4000-fold | $C_t = 0.3675 \times e^{-0.0065t}$ | 106.6 | 0.9454 |
| | | 2000-fold | $C_t = 0.5686 \times e^{-0.0054t}$ | 128.3 | 0.9716 |
| Pyraclostrobin | 25 | 2400-fold | $C_t = 0.6195 \times e^{-0.0515t}$ | 13.5 | 0.9728 |
| | | 1200-fold | $C_t = 1.5527 \times e^{-0.0481t}$ | 14.4 | 0.9332 |
| | 4 | 2400-fold | $C_t = 0.6017 \times e^{-0.0082t}$ | 84.5 | 0.9593 |
| | | 1200-fold | $C_t = 1.4310 \times e^{-0.0040t}$ | 173.3 | 0.9068 |
| Difenoconazole | 25 | 4000-fold | $C_t = 0.4221 \times e^{-0.09247t}$ | 7.5 | 0.8918 |
| | | 3000-fold | $C_t = 0.5646 \times e^{-0.03282t}$ | 21.1 | 0.8147 |
| | 4 | 4000-fold | $C_t = 0.3459 \times e^{-0.01575t}$ | 44.0 | 0.9675 |
| | | 3000-fold | $C_t = 0.6327 \times e^{-0.00513t}$ | 135.1 | 0.9291 |

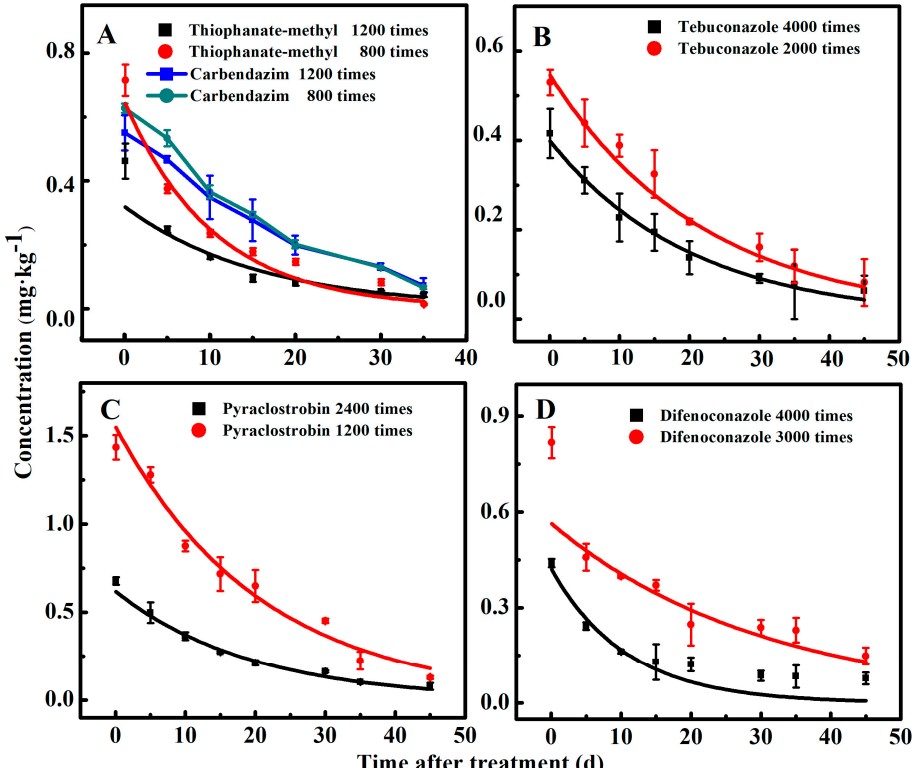

**Figure 3.** Dissipation dynamics of thiophanate-methyl and carbendazim (**A**), tebuconazole (**B**), pyraclostrobin (**C**), and difenoconazole (**D**) in pears at 25 °C.

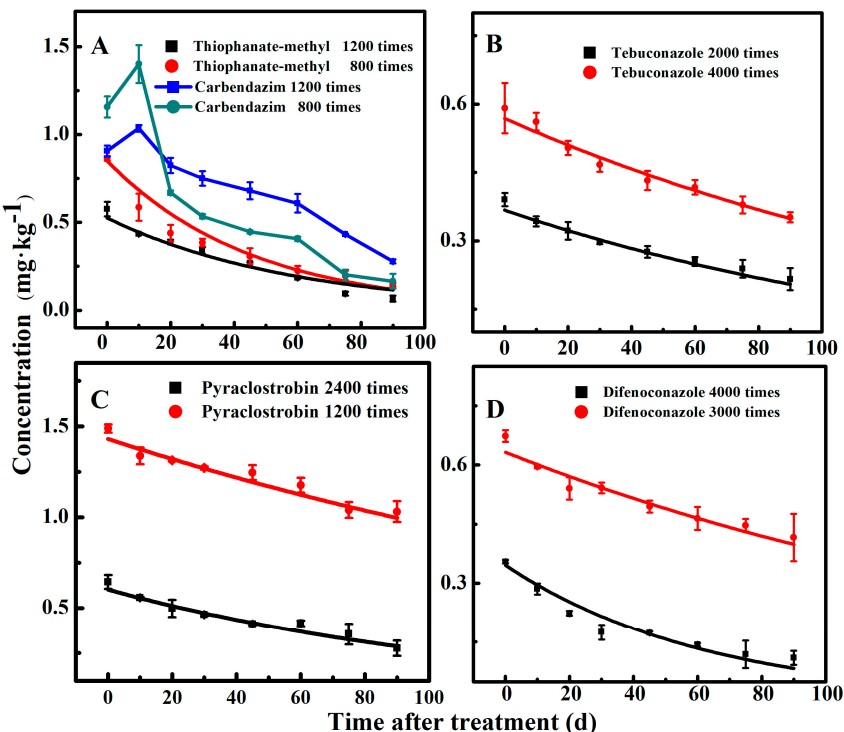

**Figure 4.** Dissipation dynamics of thiophanate-methyl and carbendazim (**A**), tebuconazole (**B**), pyraclostrobin (**C**), and difenoconazole (**D**) in pears at 4 °C.

### 3.4. Fungicide Residues of Commercial Pear Samples

To investigate chemical preservative residues in pears, pear samples were collected from producing regions and supermarkets in Anhui Province. As shown in Table S5, the positive rates of carbendazim, thiophanate-methyl, tebuconazole, pyraclostrobin, and difenoconazole in pear samples from 18 refrigerated warehouses in the producing region were 100%, 94.4%, 83.3%, 77.2%, and 72.2%, respectively. The residual concentrations of carbendazim in pear samples from 18 refrigerated warehouses ranged from 0.02 to 30.1 mg/kg, which derived from the metabolism of thiophanate-methyl or the application of carbendazim. The concentrations of thiophanate-methyl, tebuconazole, pyraclostrobin, and difenoconazole were <0.01–13.55, <0.01–0.55, 0.01–0.52, and <0.01–0.28 mg/kg, respectively. The maximum residue limits (MRLs) of thiophanate-methyl, carbendazim, tebuconazole, pyraclostrobin, and difenoconazole in pears were 3, 3, 0.5, 0.5, and 0.5 mg/kg, respectively, as set by GB 2763-2021 (the National Food Safety Standard in China). The qualified rate of pear samples from 18 refrigerated warehouses was less than 50%. Carbendazim was detected in the disqualified pear samples from refrigerated warehouses in the production region. Meanwhile, there were 10 pear samples that exceeded the MRL of thiophanate-methyl in the 18 disqualified pear samples. For tebuconazole and pyraclostrobin, only one pear sample exceeded the MRL standard. The residual concentration of difenoconazole in all the samples was lower than the MRL of 0.5 mg/kg. Therefore, the exposure risk due to pears from refrigerated warehouses in the producing region mainly resulted from carbendazim and thiophanate-methyl.

The fungicide residues of the pear samples from 24 supermarkets in Anhui Province are shown in Table S6. The residue concentrations of carbendazim, tebuconazole, thiophanate-methyl, pyraclostrobin, and difenoconazole were 0.06–5.68, <0.01–2.08, <0.01–8.44, <0.01–45.93, and 0.05–23.04 mg/kg, respectively. The positive rates of the five target compounds were 58.3–100%. Specifically, the positive rates of carbendazim and difenoconazole were 100%. The fungicide residues of the pear samples in the 24 supermarkets were different from those in the refrigerated warehouses of the producing region. The concentration of pyraclostrobin in pears from 12 supermarkets exceeded the MRL value of 0.5 mg/kg,

and the maximum residual concentration was 45.93 mg/kg. Difenoconazole residue concentrations in pear samples from 9 supermarkets exceeded their MRL (3 mg/kg). For thiophanate-methyl and carbendazim, only one and two samples were unqualified, respectively. Therefore, the exposure risk from pears in supermarkets was mainly due to pyraclostrobin and difenoconazole.

*3.5. Dietary Intake Risk of Fungicides*

To ensure the quality and safety of pears, a dietary exposure risk assessment of the fungicides was conducted according to the Guidelines for Risk Assessment of Pesticide Residues in Food of China based on the supervised trials median residue (STMR) values of commercial pear samples. The STMR values of thiophanate-methyl, tebuconazole, pyraclostrobin, and difenoconazole in pear samples are shown in Tables S5 and S6. The ADI values of tebuconazole, thiophanate-methyl, pyraclostrobin, and difenoconazole were 0.03, 0.09, 0.03, and 0.01 mg/kg bw d, respectively [36]. Considering the maximization of the health risk, the MRL values of the fungicides in different foods were selected to calculate the RQ value. As shown in Tables S7–S15, the RQ, RQa, and RQc values of the four fungicides in the pears from both the refrigerated warehouses of the producing region and the supermarkets were less than 100%, which indicates that there was a negligible chronic-term risk inherent in the consumption of pears

## 4. Discussion

In this paper, an efficient and stable analytical method was developed for the simultaneous determination of thiophanate-methyl, tebuconazole, pyraclostrobin, and difenoconazole in pears by UPLC–MS/MS. This method was successfully used to study dissipation behavior and perform a risk assessment. The results showed that the half-lives of these fungicides in pears increased 2.9–8.2-fold at 4 °C storage conditions relative to 25 °C storage conditions. In addition, a risk assessment of fungicide residues in pear samples from production areas and supermarkets was also performed. Although the flash dietary risk assessment values of RQ, RQa, and RQc for edible pears were < 1, some commercial pears in the market had excessive levels of chemical preservatives. This work provides a guideline for the risk assessment of preservatives in fruits for human health.

## 5. Conclusions

In this work, a multiresidue analysis method was established to investigate the degradation behavior of four common chemical preservatives in pear storage. The results showed that chemical preservatives persisted in pears in low-temperature storage, increasing the exposure risk of chemical preservatives. Although the intake risk assessment results showed that the chronic-term risk of consuming pear was acceptable, some commercial samples contained excessive chemical preservatives. Therefore, the government should strengthen the supervision of chemical preservatives in the market to ensure the quality and safety of agricultural products.

**Supplementary Materials:** The following supporting information can be downloaded at: https://www.mdpi.com/article/10.3390/agriculture12050630/s1, Gradient elution program of the chromatographic column (Table S1); MS parameters for the analysis of the five fungicides (Table S2); the information on meteorological conditions during sampling (Table S3); fortified recoveries of the five fungicides (Table S4); residues of the fungicides in pear samples (Tables S5 and S6); dietary intake risk assessment (Tables S7–S15).

**Author Contributions:** Conceptualization, X.W. and Z.Z.; methodology, Y.T.; software, K.H.; validation, Y.T., K.H. and X.L.; formal analysis, C.L.; investigation, Y.X.; resources, C.L.; data curation, X.L.; writing—original draft preparation, Y.T.; writing—review and editing, X.W.; visualization, Y.T.; supervision, Y.T.; project administration, Z.Z.; funding acquisition, X.W. All authors have read and agreed to the published version of the manuscript.

**Funding:** This study was supported by the Key Research and Development Program of Anhui Province (201904a06020051).

**Institutional Review Board Statement:** Not applicable.

**Informed Consent Statement:** Not applicable.

**Data Availability Statement:** The data presented in this study are available on request from the corresponding author.

**Acknowledgments:** Thanks to Taozhong Shi for his help in UPLC–MS/MS analysis.

**Conflicts of Interest:** The authors declare no conflict of interest.

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
