# Peer review of "Dissipation Dynamics and Dietary Risk Assessment of Four Fungicides as Preservatives in Pear"

_agriculture, doi:10.3390/agriculture12050630_

Round 1

Reviewer 1 Report

The study was carried out to determine the dissipation dynamics and the dietary risk assessment of four pesticides (thiophanate-methyl, tebuconazole, pyraclostrobin, and difenoconazole) used for the postharvest treatment of pears at two storage temperatures. Carbendazim was also studied as major metabolite of thiophanate-methyl.

In my opinion, this type of study is not new. In fact, there are many similar studies applied to a wide variety of crops. Moreover, only 4 pesticides were studied. On the other hand, the applied analytical method was adequately optimized (extraction solvent and SPE sorbent) and validated (n=5 at five spiked levels even at 10 ppb), and simultaneous determination of these pesticides was then performed in 24 commercial samples. Consequently, the work is well done, and the results could be of interest to the scientific community and entities interested in this crop. Then, I believe that this manuscript only needs a minor revision to correct small grammatical and typographical errors (lines 19, 86, 97, 131 etc.), as well as to check the acronyms. Finally, the format of the references is not according to the standard of the journal.

Author Response

Point 1: The study was carried out to determine the dissipation dynamics and the dietary risk assessment of four pesticides (thiophanate-methyl, tebuconazole, pyraclostrobin, and difenoconazole) used for the postharvest treatment of pears at two storage temperatures. Carbendazim was also studied as major metabolite of thiophanate-methyl.

In my opinion, this type of study is not new. In fact, there are many similar studies applied to a wide variety of crops. Moreover, only 4 pesticides were studied. On the other hand, the applied analytical method was adequately optimized (extraction solvent and SPE sorbent) and validated (n=5 at five spiked levels even at 10 ppb), and simultaneous determination of these pesticides was then performed in 24 commercial samples. Consequently, the work is well done, and the results could be of interest to the scientific community and entities interested in this crop. Then, I believe that this manuscript only needs a minor revision to correct small grammatical and typographical errors (lines 19, 86, 97, 131 etc.), as well as to check the acronyms. Finally, the format of the references is not according to the standard of the journal.

Response 1: Thanks very much for your suggestions and encouragements to our manuscript. We have carefully revised our manuscript according to your comments, including the grammatical and typographical errors, the acronyms, and the format of references (please see lines 19-20, 97-99, 129-130, 251, 265, 328-330, and 364-441). Meanwhile, the manuscript has undergone English language editing by MDPI with the experienced and native English speaking editors (English editing ID: English-43417). We have marked the sections of manuscript revision using red color and the sections of English polishing using blue color in the revised manuscript.

Reviewer 2 Report

The manuscript titled “Dissipation dynamics and dietary risk assessment of four fungicides as preservatives in pear” by Yongfeng Tang, Kuikui Hu, Xiaomeng Li, Chaogang Liu, Yanhui Xu, Zhaoxian Zhang, Xiangwei Wu describes the degradation of four fungicides in pear. In my opinion the manuscript is interesting and may be cited in the future. However in my opinion there is a lack of information about meteorological conditions during sampling. The meteorological conditions should be described (rainfall, humidity, temperature etc.).

The references need to be adjusted to the requirements.

Author Response

Point 1: The manuscript titled “Dissipation dynamics and dietary risk assessment of four fungicides as preservatives in pear” by Yongfeng Tang, Kuikui Hu, Xiaomeng Li, Chaogang Liu, Yanhui Xu, Zhaoxian Zhang, Xiangwei Wu describes the degradation of four fungicides in pear. In my opinion the manuscript is interesting and may be cited in the future. However in my opinion there is a lack of information about meteorological conditions during sampling. The meteorological conditions should be described (rainfall, humidity, temperature etc.).

Response 1: Thanks very much for your efforts and suggestions to our manuscript. We have added the imformation about meteorological conditions during sampling (Table A3) into the revised manuscript (Please see lines 135-136).

Point 2: The references need to be adjusted to the requirements.

Response 1: The format of references has been carefully rephrased according to requirements in the Author Guideline (Please see lines 364-441).

Reviewer 3 Report

English editing only

Author Response

Point 1: English editing only.

Response 1: As Reviewer #3 pointed out, we have carefully revised our manuscript, including the grammatical and typographical errors, and the acronyms. Meanwhile, the manuscript has undergone English language editing by MDPI with the experienced and native English speaking editors (English editing ID: English-43417). We have marked the sections of manuscript revision using red color and the sections of English polishing using blue color in the revised manuscript.
